# Chlorinated Persistent Organic Pollutants (PCDD/Fs and PCBs) in Loggerhead Sea Turtles Stranded along the Central Adriatic Coast

**DOI:** 10.3390/ani12223177

**Published:** 2022-11-17

**Authors:** Ludovica Di Renzo, Roberta Ceci, Silvia D’Antonio, Gabriella Di Francesco, Federica Di Giacinto, Nicola Ferri, Carla Giansante, Manuela Leva, Giulia Mariani, Vincenzo Olivieri, Simone Pulsoni, Romolo Salini, Giampiero Scortichini, Giulio Tammaro, Gianfranco Diletti

**Affiliations:** 1 Istituto Zooprofilattico Sperimentale (IZS) Dell ’Abruzzo e Molise “G. Caporale”, 64100 Teramo, Italy; 2 Centro Studi Cetacei Onlus (CSC), 65125 Pescara, Italy

**Keywords:** *Caretta caretta*, POPs, PCDD/Fs, PCBs, Adriatic Sea, sea turtles, congener pattern

## Abstract

**Simple Summary:**

Persistent organic pollutants (POPs) are a group of organic compounds characterized by long-range transport, persistence, bioaccumulation, and high toxicity. They can be found in aquatic environments where they can be ingested by marine organisms. As a long-lived, widely distributed, and opportunistic predator, the loggerhead turtle (*Caretta caretta*) is a sentinel species in the Mediterranean Sea, and it can also be considered a bioindicator of the health of the sea in terms of contaminants. In this study, 44 *C. caretta* stranded along the Adriatic coast were selected, considering sex and curved carapace length (CCL) as a proxy of age. The fat tissues and the livers of the animals were sampled and analyzed for studying the contamination level of chlorinated POPs in relation to sex and size. This work points the focus on the importance of monitoring these contaminants in *C. caretta* to evaluate the anthropic effects on the marine environment.

**Abstract:**

Persistent organic pollutants are widespread in the marine environment. They can bioaccumulate and biomagnify in marine organisms through the food web with a potentially toxic effect on living organisms. The sea turtle *Caretta caretta* is a carnivorous animal with opportunistic feeding behavior. These turtles tend to bioaccumulate pollutants through food, and hence they can be considered an indicator of chemical pollutants in the marine ecosystem. In this study, 44 loggerhead sea turtles were considered, and liver and fat tissue were sampled from each of them to investigate the levels of dioxins (PCDD/Fs) and polychlorinated biphenyls (PCBs) in sea turtles and their potential correlation with sex and size in terms of curved carapace length (CCL). Results suggested that these contaminants were easily bioaccumulated, and PCBs were predominant compared to dioxins in both liver and fat tissue. The congener patterns were similar to those found in sea fish. Moreover, there were no differences in the contamination levels between females and males, nor was there a correlation with the size. There is a need to harmonize the methodological approaches to better evaluate the results and trends over time and to monitor the species and indirectly the health status of the marine environment.

## 1. Introduction

Persistent organic pollutants (POPs) comprise a group of ubiquitous environmental contaminants. They are considered priority pollutants because of their toxicity to living organisms, their persistence in the environment, and their bioaccumulation through the food web [1,2,3]. POPs include a large variety of halogenated compounds. Among them, the dioxin group generally refers to polychlorinated dibenzo-p-dioxins (PCDDs) and dibenzofurans (PCDFs), namely, 75 and 135 congeners, respectively. Furthermore, dioxin-like polychlorinated biphenyls (DL-PCBs) are also included in the “dioxins” group [4]. Dioxins can be produced as a consequence of several industrial processes (smelting, chlorine bleaching of paper pulp, and the manufacturing of some herbicides and pesticides) but also volcanic eruptions and forest fires [5,6]. On the contrary, PCBs were mainly used in the past as lubricants by the power industry in electrical transformers, capacitors, and hydraulic equipment [5,6]. POPs can reach the aquatic environment both directly through, for example, transformer spills and agricultural runoff, or indirectly via atmospheric transport and ocean currents [5]. Once in the marine environment, they bioaccumulate in marine organisms, especially in fatty tissues, due to their lipophilic nature, and they bio-magnify along the food chain, playing a toxic role [6,7]. Several studies have reported high levels of dioxins and PCBs in dolphins [8], seals [9], sharks [10], predator birds [11], and human tissues [12]. Several health issues such as dysfunctional immune systems, endocrine disruption, cardiovascular diseases, cancers, diabetes, dysfunctional reproductive systems, and birth defects in humans and wildlife can be caused by their exposure to these contaminants [2,13,14].

The bioaccumulation of POPs and the effects on marine life have become matters of concern for their transfer along the food chain and their impact on wildlife species [15,16,17,18,19]. In this regard, it is important to monitor the fate of these contaminants in the marine environment to improve the effectiveness of conservation strategies for sea health and vulnerable species [20,21]. In 2008, the European Commission drafted a Marine Strategy Framework Directive (MSFD) with the intention of reaching Good Environmental Status (GES) for the European marine environment by 2020 [22,23]. The MSFD remarked on the importance of monitoring the presence of chemical contaminants in the marine environment. In this context, loggerhead sea turtles have been widely used as bioindicators because of their longevity and migratory behavior [21]. Sea turtles are long-lived marine reptiles and are widespread worldwide, from coastal to oceanic/pelagic habitats [24]. They are opportunistic and carnivorous predators and are believed to bioaccumulate pollutants from food and from the surrounding environment, including sediment, water, and plastic debris. Some papers reported that the levels of chlorobisphenyls (CBs) and organochlorine pesticides (OCPs) are higher in tissues of loggerhead sea turtles than in tissues of green and leatherback turtles, and this may be due to the different dietary habits [25,26,27]. As they are widely distributed from oceanic to neritic habitats, and they are opportunistic predators feeding mostly on benthic organisms [28], loggerhead sea turtles can be also considered bioindicators of chemical pollution in marine ecosystems [29,30,31,32]. The *C. caretta* is the most common species of sea turtle in the Mediterranean Sea, a semi-closed basin, with highly anthropized coasts [33]. For these reasons, the environmental pollution and persistence of chemical contaminants are expected to be more critical compared to other geographic areas [34].

Several matrices have been used for ecotoxicological studies in sea turtles; liver and fat are usually the most analyzed matrices for assessing the chemical and physical properties of POPs [29]. The overview by D’Ilio et al. (2011) [29] highlighted that the level of POPs is 10 times higher in the liver of sea turtles from the Atlantic Ocean than the Mediterranean Sea. The levels of contamination in blood and plasma seem to be lower than in other tissues, which may be due to the lower solubility of POPs in blood. Recently, several studies have focused on chlorinated contaminants such as PCBs or pesticides, while only a few data are available on dioxins. Moreover, the studies conducted so far employed different matrices, making comparisons difficult to infer.

This study aimed to quantify PCDD/Fs and DL- and NDL-PCBs in sea turtles stranded along the coast of the Central Adriatic Sea using isotope dilution coupled with high resolution mass spectrometry. The occurrence data were used to investigate the correlation between the level of contaminants in loggerhead sea turtles and the size and the sex of the animals, considering only healthy individuals. Regarding the size, we hypothesized a positive correlation between the size/age of the animals (in terms of CCL) and the levels of contaminants that are believed to accumulate in an age-related manner, as they are lipophilic [20,30]. Esposito et al. (2022) [21] highlighted a weak positive correlation between size and PCB concentration, with adult females showing higher levels compared to juveniles. On the contrary, a study by Casini et al. (2018) [20] did not find any correlation between the levels of PCBs in blood and the size (CCL), even though the mean values were higher in larger sized sea turtles. Regarding sex, we hypothesized that reproductive females could reduce their contamination, offloading a significant portion of the POPs burden into their eggs. On the other hand, males do not have a similar offloading mechanism, continuing to accumulate POPs.

The data obtained in this work can contribute to providing information about the environmental quality of the habitat of *C. caretta.*

## 2. Materials and Methods

### 2.1. Sample Collection

The regional networks of intervention for stranded marine animals (sea turtles included) of Abruzzo (DG 2014 21/167) and Molise (DCA n.67 of 22 May 2019) regions are in charge of monitoring the coast under their jurisdiction [35,36]. According to the cadaveric alteration of the carcasses, when they were not in an advanced state of decomposition, they were transferred to the laboratory at Istituto Zooprofilattico Sperimentale dell’Abruzzo e del Molise (IZS-Teramo) to define the cause of death. Diagnostic examinations were conducted at IZS-Teramo, such as external inspection (biometrics and sex identification); complete necropsy examination; and bacterial, virological, and parasitic investigations. According to Poppi and Marchiori [37], the body condition at necropsy was classified as poor, fair, good, or excellent based on the appearance of muscular and fat tissue in the inguinal region, and the sex was finally determined by a visual examination of gross gonadal morphology. In addition, the content of the digestive tract was evaluated for Good Environmental Status in terms of the presence of litter ingested according to Marine Framework Strategy Directive (MFSD), and the assessment of dietary habits.

To evaluate the level of POPs, liver and fat tissue of each carcass were collected during necropsy, packed in aluminum foil, and stored at −20 °C. Turtles were included in the study design unless both the fat tissue and the liver were available. Pools were defined based on sex (female and male), sexual maturity (mature and immature), and the curved carapace length (CCL). An interval of 10 cm was considered for the pools to better understand how the level of contaminants varies with the size. The 5 classifications considered were: (1) from 31 to 40 cm, (2) from 41 to 50 cm, (3) from 51 to 60 cm, (4) from 61 to 70 cm, and (5) from 71 to 80 cm. A total of 44 sea turtles with a CCL ranging from 33 to 77 cm was selected for chemical investigations. On the basis of size and sexual maturity, 14 pools were created for both liver and fat tissue.

### 2.2. Chemical Analysis

#### 2.2.1. Chemicals and Analytical Standards

Solvents (toluene, dichloromethane, isooctane, acetone, n-hexane, and ethanol) and other reagents such as sulphuric acid, ammonia solution, anhydrous sodium sulphate, and sodium chloride were purchased as analytical grade from Honeywell Burdick & Jackson in Seelze, Germany. Ultra-pure water was generated through a Purelab option-Q system (ELGA, High Wycombe, Bucks, UK).

All standards were purchased from Wellington Laboratories Inc. Ontario, Canada. Calibration solutions DF-CVS-C10 (CS1 through CS4), ^13^C_12_-labelled internal standard DF-LCS-C200, and recovery standard DF-IS-J were used for PCDD/F analysis. Calibration solutions WP-CVS (CS1 through CS7), ^13^C_12_-labelled internal standard WP-LCS, and recovery standard P48-RS-STK were used for DL-PCB analysis. Calibration solutions P48-M-CVS (CS1 through CS5), ^13^C_12_-labelled internal standard P48-M-ES, and recovery standard P48-RS-STK were selected for NDL-PCB analysis.

#### 2.2.2. Sample Preparation

Each pool was homogenized by a knife mill Grindomix GM-200 (Retsch, Dusseldorf, Germany) before analysis. Samples were analyzed for PCDD/Fs and PCBs using accredited methods following ISO EN17025, based on US EPA (1994) Method 1613 B for PCDD/Fs and US EPA (2008) 1668 B for PCBs. The analytical methodology, based on isotope dilution, was widely described by Diletti et al. (2007) [38]. Briefly, an aliquot of the homogenized sample (~3 g for fat tissue and ~5 g for liver) was fortified with the labeled standard solutions for PCDD/Fs and PCBs (0.2–0.4 ng of PCDD/Fs, 1.0 ng of DL-PCBs, 2.0 ng of NDL-PCBs). All the samples were mixed with anhydrous sodium sulphate at a ratio between one-third to half of the sample weight and left to equilibrate for 6–12 h. Samples were extracted using an ASE 350 Thermo Scientific Dionex (Sunnyvale, CA, USA) extractor. Three extraction cycles were performed at 125 °C and 1500 psi using a mixture of n-hexane and acetone 80:20 (*v*/*v*). Extracts were filtered through anhydrous sodium sulphate, and the extraction solvent was removed with a rotary evaporator. The lipid content was determined by gravimetric determination. After solubilization in hexane, the extracts were subjected to liquid–liquid partitioning with concentrated sulphuric acid, 20% aqueous potassium hydroxide, and saturated aqueous sodium chloride. The sample volumes were reduced to approximately 1 mL by a rotary evaporator and then purified on an automated Power-Prep™ system (Fluid Management System (FMS), Billerica, MA, USA) using multilayer silica, activated carbon, and alumina columns (obtained directly from FMS). The two eluates were dried under a gentle stream of nitrogen and re-dissolved in 20 μL of the respective recovery (syringe) standard solutions.

#### 2.2.3. Instrumental Analysis

The analyses were carried out using high-resolution gas chromatography/high-resolution mass spectrometry (HRGC-HRMS). Compounds were analyzed using a GC Trace Series 2000 coupled to a MAT 95 XL (Thermo Fisher Scientific, Waltham, MA. USA) for PCBs and a Trace Series 1310 GC coupled to a DFS (Thermo Fisher Scientific, USA) for PCDD/Fs.

PCDD/F congeners were separated on a 60 m × 0.25 mm (0.10 μm film thickness) DB-5 MS capillary column (J&W Scientific, Folsom, CA, USA), while the separation for PCBs was obtained using an HT-8 capillary column (SGE Analytical Science, Victoria, Australia). A volume of 1 μL of sample extracts was injected into a split/splitless injector held at 280 °C using helium (1 mL/min) as a carrier gas. The GC/MS interface was set to 280 °C. The chromatographic programs were described before by Ceci et al [39].

PCBs: 120 °C for 0.5 min, ramped to 180 °C at 20 °C min^−1^, ramped to 260 °C at 2 °C min^−1^, ramped to 300 °C at 5 °C min^−1^, ramped to 310 °C at 1.6 °C min^−1^.

PCDD/Fs: 120 °C for 2 min, ramped to 220 °C at 10 °C min^−1^, held at 220 °C for 11 min, ramped to 235 °C at 3 °C min^−1^, held at 235 °C for 5 min, ramped to 315 °C at 3.5 °C min^−1^.

The mass spectrometer was tuned to a resolution of 10,000 using perfluorotributylamine (FC-43). Mass filtering was carried out with electron ionization (EI) at 50 eV in the selected ion monitoring (SIM) mode. Further details of the conditions used for monitoring have been reported for individual analytes [38,40].

Seventeen 2,3,7,8-chloro substituted PCDD/Fs (2,3,7,8-TeCDD, 1,2,3,7,8-PeCDD, 1,2,3,4,7,8-HxCDD, 1,2,3,6,7,8-HxCDD, 1,2,3,7,8,9-HxCDD, 1,2,3,4,6,7,8-HpCDD, OCDD, 2,3,7,8-TCDF, 1,2,3,7,8-PeCDF, 2,3,4,7,8-PeCDF, 1,2,3,4,7,8-HxCDF, 1,2,3,6,7,8-HxCDF, 1,2,3,7,8,9-HxCDF, 2,3,4,6,7,8-HxCDF, 1,2,3,4,6,7,8-HpCDF 1,2,3,4,7,8,9-HpCDF, OCDF), twelve dioxin-like PCBs (CB-77, CB-81, CB-126, CB-169, CB-105, CB-114, CB-118, CB-123, CB-156, CB-157, CB-167, CB-189), and six NDL-PCBs (CB-28, CB-52, CB-101, CB-138, CB-153, CB-180) were detected. The sums of PCDD/Fs and DL-PCBs were reported in Toxic Equivalents (WHO-TEQs), where the concentration of each congener is multiplied by its Toxic Equivalency Factor (TEF) [41]. Results for NDL-PCBs were reported as the sum of six indicator congeners. Results were reported in Upper Bound, and all values below the limit of quantification (LOQ) were set to be equal to the respective LOQ. Results were reported on a wet weight (ww) basis for liver samples and on a lipid basis (lb) for fat tissues.

#### 2.2.4. Quality Control

A procedural blank was added to each batch consisting of 10 samples, and a calibration curve for individual congeners was systematically verified for linearity of response. The LOQ for the toxicological sums was about 0.02 pg WHO-TEQ g^−1^ for PCDD/Fs, 0.05 pg WHO-TEQ g^−1^ for DL-PCB, and 0.02 ng g^−1^ for the analytical sum of NDL-PCBs. Samples were analyzed according to the requirements for food reported in the EU Commission Regulation (EC) No 644/2017 [42]. The EU Regulation requires the use of HRMS; the use of ^13^C-labeled surrogates/internal standards; the check of the recovery of labeled standards; specifications on sensitivity, accuracy, and precision; and the measurement of uncertainty. The reliability of results was ensured by the participation in several proficiency tests on food organized by the European Union Reference Laboratory for Halogenated Persistent Organic Pollutants in Feed and Food over fifteen years, obtaining satisfactory results (z-scores < 2.0). The measurement uncertainty was in the order of 18% for WHO-TEQs and the sum of six NDL-PCBs.

### 2.3. Statistical Analysis

The data were analyzed using R software [43]. The Mann–Whitney test was used to compare the contamination levels in liver and fat tissues between males and females. Furthermore, the Kruskall–Wallis test was performed to compare the contamination levels in liver and fat tissues among the five size-specific groups. The significant level (*p*-value) was set to 0.05.

## 3. Results

### 3.1. Sample Collection

From March 2017 to December 2018, the Regional Marine Animals Standing Network of the Abruzzo and Molise regions intervened on 160 sea turtles that were stranded and dead. A total of 76 carcasses was transferred to the IZS-Teramo for the necroscopic examination. Among them, 44 sea turtles found in good/optimum body condition and were selected for chemical investigations. Sea turtles’ CCLs ranged from 33.0 to 77.0 cm.

According to macroscopic evaluation of the sexual organs, among the 27 females, 17 were determined as immature and 10 as mature; among the 17 males, 15 were immature, while 2 were mature. According to the size and sexual maturity, 14 pools were formed as summarized in Table 1.

### 3.2. Contamination Levels

In this study, 17 PCDD/Fs, 12 DL-PCBs, and 6 NDL-PCBs were investigated in the 14 pools of samples from loggerhead sea turtles, both for liver and fat tissue. PCDD/F and DL-PCB toxicological sums and the analytical sum of NDL-PCBs detected for both of the matrices in each pool are reported in Table 2 and Table 3. The lipid content is also reported, and it ranged between 1.5 and 8.9% in liver and between 21.6 and 91.4% in fat tissue.

PCDD/Fs ranged from 0.20 to 0.98 pg WHO-TEQ g^−1^ in liver and from 0.73 to 6.1 pg WHO-TEQ g^−1^ in fat tissue. Moreover, PCB levels were higher than PCDD/Fs in all samples analyzed. The DL-PCB toxicological sums ranged between 0.76 and 3.2 pg WHO-TEQ g^−1^ in liver, while in fat tissue were found values ranging from a minimum of 4.6 to a maximum of 16 pg WHO-TEQ g^−1^. NDL-PCB values had great variability in both matrices. They ranged between 3.8 and 40 ng g^−1^ in liver and from 44 ng g^−1^ to 140 ng g^−1^ in fat tissue.

Concentrations of PCDD/F congeners ranged from 0.05 to 0.64 pg g^−1^ in liver and from 0.08 to 3.9 pg g^−1^ lipid basis in fat tissue. Among the 17 PCDD/Fs investigated, the majority was quantified, and a higher relative abundance of furans compared to dioxins was observed in both tissues. The most abundant congeners were 2,3,7,8-TCDF, 2,3,4,7,8-PeCDF, and 1,2,3,7,8-PeCDF. For dioxins, 12378-PeCDD and OCDD were detected at similar concentrations, followed by 2,3,7,8-TCDD and 1,2,3,6,7,8-HxCDD (PCDD/F pattern is reported in Figure 1). Only two congeners were always under LOQ, namely, 1,2,3,7,8,9-HxCDF and 1,2,3,4,7,8,9-HpCDF. The main contributors to the toxicological sum were PeCDD (37% and 31%) and PeCDF (26% and 29%), respectively, for liver and fat. These two congeners contributed to more than 60% of the toxicological sum. The toxicological sum for PCDD/Fs, expressed as mean ± standard deviation, was 0.59 ± 0.26 for liver and 2.9 ± 1.6 pg WHO-TEQ g^−1^ lb for fat tissue.

Concentrations of DL-PCB congeners ranged from 1.6 to 2301 pg g^−1^ ww in liver and from 11 to 12,050 pg g^−1^ lb in fat tissue. Among the 12 DL-PCBs, all congeners were quantified, and PCB-118, PCB-105, and PCB-156 were the most abundant. PCB-126 made the largest contribution to the toxicological sum (>88%). The mean values of the toxicological sums of DL-PCBs were 1.8 ± 0.72 pg WHO-TEQ g^−1^ ww for liver and 11 ± 2.9 pg WHO-TEQ g^−1^ lb for fat tissue. The DL-PCBs pattern is reported in Figure 1.

Concentrations of NDL-PCB congeners ranged from 0.02 to 7.6 ng g^−1^ in liver and from 0.10 to 43 ng g^−1^ in fat tissue. All of six NDL-PCBs were detected, and PCB-153 was the most abundant, followed by PCB-138 and PCB-180. Averages of the NDL-PCB analytical sum were 15 ± 10 ng g^−1^ for liver and 84 ± 31 ng g^−1^ for fat tissue. The NDL-PCB pattern is reported in Figure 1.

### 3.3. Statistical Analysis

The results of Mann–Whitney and Kruskall–Wallis tests indicated no differences in contamination levels between males and females (Figure 2 and Appendix A) or among the five size-specific groups (Figure 3 and Appendix A).

## 4. Discussion

In this study, 44 samples each of liver and fat from loggerhead sea turtles (*C. caretta*) in good/optimum body condition were considered. For each matrix, 14 pools were defined on the basis of the characteristics described before, and PCDD/Fs and DL- and NDL-PCBs were analyzed.

Just a few comparisons could be performed because of the limited data on *C. caretta* available in the literature, in particular on PCDD/Fs contamination [44,45]. Conversely, more studies focused on other POPs [29,46,47,48]. However, these studies were very heterogeneous regarding the analyzed matrices (liver, muscle, blood, or fat tissue), the classes of investigated contaminants (from organochlorine pesticides to PCBs or PBDEs), and the expression of results (wet weight, lipid weight, or WHO-TEQ) [49]. Moreover, since little is known about the levels of contaminants in sea turtles, sometimes different congeners from the same group of POPs were examined, and biometric characteristics, sex, and sexual maturity were not always specified, as well as the evaluation of the nutritional status.

Regarding PCDD/Fs contamination, few data are available on *C. caretta*. In a study conducted by Lambiase et al. (2021) [45] on the livers of 24 stranded sea turtles, a mean value of 3.6 ± 2.5 pg WHO-TEQ g^−1^ ww was found. This value was higher than the PCDD/Fs toxicological sums detected in this study, which reached a maximum value of 0.98 pg WHO-TEQ g^−1^ in liver (Table 2) with a mean value of 0.59 ± 0.26 pg WHO-TEQ g^−1^ ww. More data are available on PCBs, both for DL- and NDL-PCBs. In the study of Lambiase et al. (2021) [45], a mean value of 8.7 ± 9.0 pg WHO-TEQ g^−1^ ww for DL-PCBs was obtained as a toxicological sum. A lower value was found in this work with a mean value of 1.8 ± 0.7 pg WHO-TEQ g^−1^ ww in the liver. The livers of 12 sea turtles were analyzed by Storelli et al. (2014) [44] for PCDD/Fs and DL-PCBs, and a toxicological sum of 1.50 pg WHO-TEQ g^−1^ ww was reported. This value was in the same order of magnitude as those found in the present study, where the sum of PCDD/Fs and DL-PCBs in liver was 2.3 ± 1.0 pg WHO-TEQ g^−1^ ww. Few studies have been conducted on the fat tissues of *C. caretta* sea turtles. Lazar et al. (2011) [47] and Cammilleri et al. (2017) [48] analyzed the six NDL-PCBs in stranded loggerhead turtles and found very high contamination levels in adipose tissue. The analytical sum of NDL-PCBs found in this study was 84 ± 30 ng g^−1^ lb, about one order of magnitude lower than that cited above.

The differences observed in the comparisons between the obtained results and those found in the literature can be ascribed to several variables belonging to the biology of the species under study, the different nutritional status, the areas of foraging, and sex and reproduction status. In addition, it is also important to consider the performances of the different methods used for chemical analysis in terms of equipment (electron capture detector, low- or high-resolution mass spectrometry), limits of quantification, and use of ^13^C_12_-labelled internal standard.

As usually observed in fish [50,51], in this study, DL-PCB levels were about three times more abundant than PCDD/Fs, both in the liver and fat tissue. In particular, the higher chlorinated PCB congeners— hexa- and hepta-chlorobiphenyls—were predominant, as also reported by Storelli et al. (2014) [44] and Lazar et al. (2011) [47]. The obtained pattern could reflect the chemical composition of old commercial formulations containing PCB mixtures possibly dispersed in the aquatic environment, accumulating in sediments and fish. Sea turtles, like other predators, bioaccumulate POPs because they are at the top of the marine food chain, feeding on jellyfish, shellfish, and small fish. The contaminant transfer is also reflected in the different levels of dioxin congeners. Overall the PCDD/F pattern is comparable to that typically found in fish, where 2,3,7,8-TCDF, 2,3,4,7,8-PeCDF, 1,2,3,7,8-PeCDD, and 1,2,3,7,8-PeCDF represent the most abundant congeners [52,53]. This pattern was similar to those found in a few studies reporting PCDD/F levels [44,45]. As suggested by Storelli et al. (2014) [44], lower chlorinated compounds could easily permeate cellular membranes because of their small molecular size. Moreover, 2,3,7,8,chloro-substituted compounds are hardly metabolized and easily bioaccumulated [44]. In this study, the toxicological sum was almost six times higher in fat tissue than in liver for PCDD/Fs and about seven times for DL-PCBs and NDL-PCBs, However, it should be taken into account that the contamination levels in liver are expressed in wet weight but in fat tissue on a lipid basis.

Further investigation assessing the different contamination levels between male and female sea turtles showed no significant differences in contamination levels, with p-values always greater than 0.05 (Figure 2). This was in accordance with most of the literature. In fact, two studies, from Lazar et al. (2011) [47] and Cammilleri et al. (2017) [48], both supported that there are no differences between the sexes.

Therefore, all the sea turtles (14 pools) were considered as a unique population, and a positive correlation between contaminant levels and CCLs was investigated. It was expected that contamination levels could increase with CCL levels as a proxy of the turtle’s age. Some authors found a weak correlation between turtle size and contamination levels, and in most cases only regarding a few congeners (PCB-81 [45], PCB-52, and PCB-114 [47]), assuming that the long-term exposure was responsible for the increase of some PCB congeners. On the other hand, Mckenzie et al. (1999) [27] found a negative correlation between size and contamination levels. This could be partially explained by the effect of growth dilution. Otherwise, the data obtained in this study did not show any correlation between size and contamination levels, as confirmed by the statistical analysis (Figure 3).

The absence of any correlation could be due to the pool approach instead of single samples analysis. This resulted in an elevated dispersion of data and decreased the power of the statistical analysis. However the pools were generated to reach a reasonable compromise between analytical costs and biometric parameters.

Further investigations should be conducted to obtain a more robust statistical analysis, increasing, if possible, the sample size in order to obtain more detailed data and to be able to assess other correlations.

## 5. Conclusions

This study assessed the contamination levels of PCDD/Fs and PCBs in the liver and fat tissues of a selected group of *C. caretta*, with a good status of nutrition, stranded along Central Adriatic coast. These contaminants were found in both liver and fat tissues, corroborating the evidence that these compounds are easily bioaccumulated in sea turtles. Among the investigated congeners, PCBs were predominant compared to PCDD/Fs. Moreover, no differences in contamination levels between females and males have been found, nor have differences concerning size been found.

It would be advisable to harmonize the studies by using common and shared methodological approaches to better compare the results, evaluate the contamination trend over time, and comprehend the mechanisms of biomagnification in sea turtles.

In this way, it would be possible to safeguard the health status of sea turtles and indirectly monitor the health status of the marine environment.

## Figures and Tables

**Figure 1 animals-12-03177-f001:**
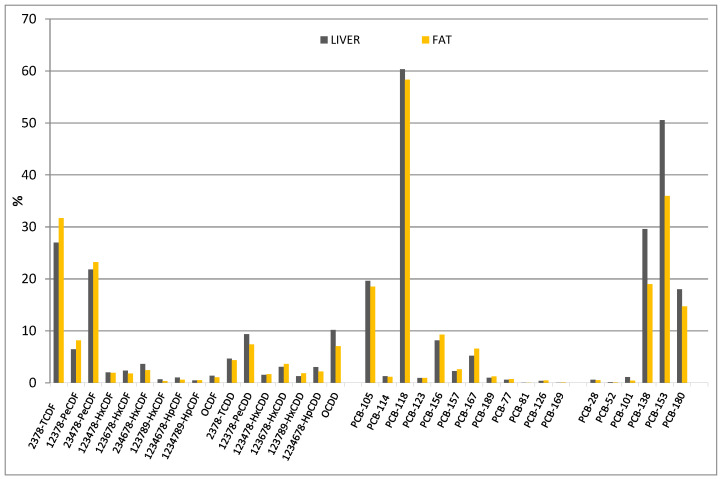
PCDD/Fs-, DL-PCBs-, and NDL-PCBs-normalized congener patterns in liver and fat tissue. Each pattern was normalized for the sum of the congeners of the respective class of contaminants.

**Figure 2 animals-12-03177-f002:**
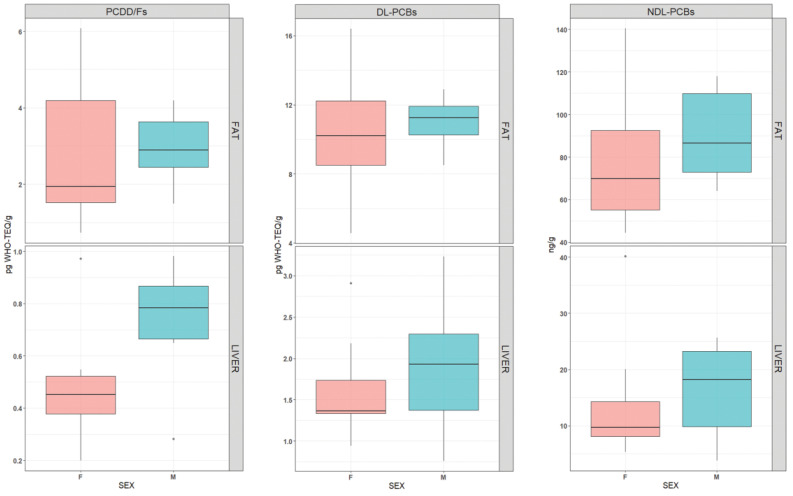
Box-plot showing the comparison of the PCDD/Fs, DL-PCBs (pg WHO-TEQ g−1), and NDL-PCBs (ng g^−1^) between male (M -cyan) and female (F–pale red) in fat tissue (on the top) and liver (on the bottom) of sea turtles.

**Figure 3 animals-12-03177-f003:**
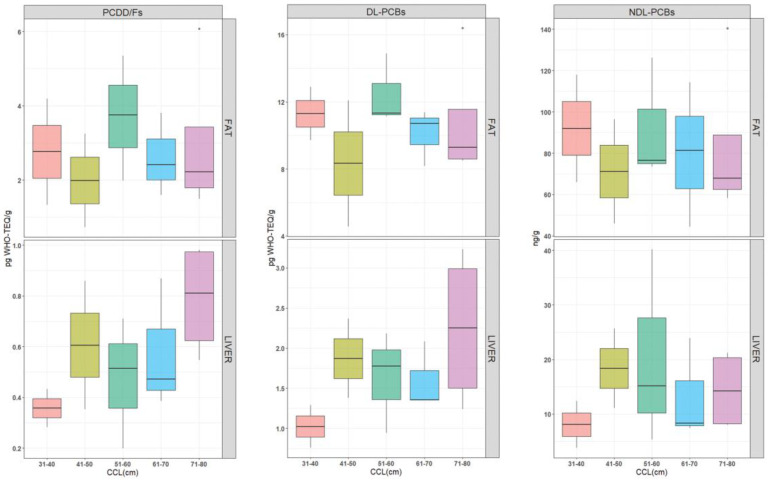
Box-plot showing the levels of PCDD/Fs, DL-PCBs (pg WHO-TEQ g-1), and NDL-PCBs (ng g^−1^) in sea turtles sorted by CCL range in fat tissue (on the top) and liver (on the bottom). From left to right: 31–40 cm (pale red), 41–50 cm (olive green), 51–60 cm (green), 61–70 cm (cyan), and 71–80 cm (pink).

**Table 1 animals-12-03177-t001:** Variables considered in pool construction.

Pool	Number of Carcasses	Sexual Maturity	CCL Range (cm)
1	5	iF	31–40
2	1	iF	41–50
3	8	iF	51–60
4	2	F	51–60
5	2	iF	61–70
6	5	F	61–70
7	1	iF	71–80
8	3	F	71–80
9	5	iM	31–40
10	3	iM	41–50
11	4	iM	51–60
12	2	iM	61–70
13	1	iM	71–80
14	2	M	71–80

F = Female, M = Male, iF = Immature Female, iM = Immature Male.

**Table 2 animals-12-03177-t002:** PCDD/Fs and PCBs levels in liver. Results are expressed as upper-bound on wet weight.

	Pool	Lipid	PCDD/Fs	DL-PCBs	NDL-PCBs
		%	pg WHO-TEQ g^−1^	pg WHO-TEQ g^−1^	ng g^−1^
Female	1	2.8	0.43	1.3	12
	2	1.6	0.35	1.4	11
	3	1.7	0.51	2.2	40
	4	3.4	0.20	0.94	5.3
	5	8.9	0.47	1.4	7.4
	6	3.5	0.39	1.4	8.3
	7	8.1	0.55	1.6	8.3
	8	2.7	0.97	2.9	20
	9	1.5	0.28	0.76	3.8
	10	1.8	0.86	2.4	26
Male	11	3.5	0.71	1.8	15
	12	8.2	0.87	2.1	24
	13	1.8	0.65	1.2	8.0
	14	3.3	0.98	3.2	21

**Table 3 animals-12-03177-t003:** PCDD/Fs and PCBs levels in fat tissue. Results are expressed as upper-bound on lipid basis.

	Pool	Lipid	PCDD/Fs	DL-PCBs	NDL-PCBs
		%	pg WHO-TEQ g^−1^	pg WHO-TEQ g^−1^	ng g^−1^
Female	1	52	1.3	9.7	66
	2	22	0.73	4.6	46
	3	72	5.4	15	126
	4	68	2.0	11	73
	5	91	1.6	8.2	44
	6	86	3.8	12	81
	7	61	1.9	8.6	58
	8	88	6.1	16	140
	9	44	4.2	13	118
	10	63	3.3	12	96
Male	11	64	3.8	11	77
	12	72	2.4	11	114
	13	68	1.5	8.5	64
	14	87	2.6	9.9	72

## Data Availability

Not applicable.

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
