# Peer review of "Chlorinated Persistent Organic Pollutants (PCDD/Fs and PCBs) in Loggerhead Sea Turtles Stranded along the Central Adriatic Coast"

_animals, 2022, doi:10.3390/ani12223177_

Round 1
Reviewer 1 Report
This is a potentially interesting manuscript but I have some major concerns about manuscript focus, writing style, methods and statistical analyses. Please see below for further details.
Lines 49-53: “Several studies have reported high levels of dioxins in dolphins, seals, sharks, predator birds, and human tissues [11]. Exposure of humans and wildlife to these contaminants may cause health problems such as endocrine disruption, cardiovascular diseases, cancers, diabetes, birth defects, and dysfunctional immune and reproductive systems [2]”. I believe such statements should be based on more references. I am confident the Authors can find further references without problems.
Why Caretta caretta can be considered a suitable sentinel species? Some justification should be provided.
Lines 65-66 and 80-82: “studies but still little is known about the effects of POPs on this animal [6, 19]”. And also: “At present, the correlation between the level of POPs contamination and the effects on sea turtles’ health is still under-investigated and poorly understood due to the limited 81 and heterogeneous data”. I can agree, but not even the present manuscript focuses on this issue, so I would avoid this statement in the Introduction, because it could induce the reader to believe that the study will investigate the effects of POPs on this species.
Why some samples collected on the coasts of two Italian regions should be representative of the whole Adriatic Sea? I am not saying that I cannot agree with this statement, but it need to be justified by citing the suitable literature. If not, the title and geographical focus should be modified accordingly.
Lines 84-86: “In detail, 17 PCDD/Fs, 12 DL-PCBs, and 6 NDL-PCBs were detected in the liver and fat tissues of C. caretta stranded along the coast of the Abruzzo and Molise regions”. These are Results, I cannot understand why they are inserted here.
Lines 84-86: “The occurrence data was used to investigate the correlation between the level of contaminants in loggerhead sea turtles, the sex, and size of the animals”. It would be nice to read some hypotheses/predictions about these objectives.
Lines 100-102 and 110-111: “Body condition at necropsy was classified as poor, fair, good, or excellent based on the appearance of muscular and fat tissue in the inguinal region, […] For the latter, 5 classifications were considered: (1) from 30 to 40 cm, (2) from 41 to 50 cm, (3) from 51 to 60 cm, (4) from 61 to 70 cm, (5) from 71 to 80 cm”. These categorizations seem to be completely arbitrary. They should be justified by citing suitable literature, or at least the rationale of these decisions should be provided, explaining all the details in an Appendix.
Why two different statistical analyses for the two study questions? This kind of analyses increases the probability of having statistically significant results. The correct way to perform these analyses is by considering together both study questions by implementing GLMs or GLMMs.
There are too many figures, please combine some graphs in the same figure.
The graphs in the last two figures are impossible to be read.
Author Response
Dear Reviewer, would you please find attached the point by point answers to your comments.
Best regards

Reviewer 2 Report
Dear Editor and Authors,
the paper is very interesting and it deals with a current issue that needs enhancement. I think the Introduction should be a bit harmonized. I'll explain. There are all the necessary elements but the logical thread breaks in some places or it could be better proposed.
There is no contextualization of the problem within European policies. Marine Strategy is mentioned but it is not clear how the species is "used" in this framework. This aspect can be better explained as it adds value to the work, in my opinion.
With regard to statistical analysis, I would like to better understand the choice of one or both tests on the dataset that is not clear to me. In any case, the results of the analysis must also be reported in a Table not only as a boxplot. Certainly the boxplot is intuitive but there really is so much variability in the samples that it sounds strange that there is not a significant difference in the samples by sex and size. Certainly the fact that there are a limited number of samples per CCL class can affect this result. It would also be interesting to check possible changes in concentration as a function of the stage of maturity but I realize that with the few samples taken it is not feasible but it is a starting point for the future.
Finally, I suggest to deepen the bibliographic analysis because there are a number of paper that deal with the subject and could help in the intro and in the discussions.

Author Response

(The authors gave the same response as above.)

Reviewer 3 Report
The manuscript presents several problems, mainly an anticipated correlation between the presence of persistent pollutants in the animals under study and a hypothetical state of health of the marine environment, which is not further described or discussed. Without the "bioindicator" element and/or any correlation between the possible presence of pollutants and some alteration in the health of the animals, all the work is reduced to the simple dosage of pollutants already known as capable of bioaccumulation and biomagnification, which adds nothing to the current state of knowledge on the subject and in some points suffers from approximation and lack of scientific rigor. Furthermore,
14-17: simple summary does not mean restricted summary, but suitable for a lay reader.
23-24: are the 88 samples related to the 44 turtles indicated in the simple summary? data have to be standardized
17 and 24: "PCDD / Fs, DL-PCB and NDL-PCBs" acronyms must be explained the first time they are mentioned
36-37: the sentence is redundant and full of repetitions
39-44 and 380: Citation 3 is no longer available on the web. At the same time, there are more specialized scientific sources from which to draw general information on organochlorine pollutants
45-53: the level of the discussion is really basic; it could have been written by a student
56-57: the sentence is redundant and full of repetitions
70-71: what do the authors mean by “They tend to bioaccumulate pollutants from food as well as from sediment”? loggerhead sea turtles are not detritivores
71-72: the correlation between eating benthic organisms and being an indicator of chemical pollution is not clear…
86-87: What should be the correlation between a persistent pollutant and the sex of animals exposed?
88-89: sea turtles are not sessile organisms, having stranded on the Adriatic coasts does not mean to be representative of the state of pollution of this basin
97-98: How can the cause of death of a sea turtle monitor the health status of the marine environment? Fishing hooks and lines, driftnets, boat strikes, etc. are common causes of death for sea turtles
201-204: these are “Materials and Methods”, and should be in section 2.1
Table 1: How is it possible that animals of 70-80 cm CCL are still sexually immature?!? Did the authors consider the season of the stranding in relation to an eventual reproductive activity?
212-218 and Table 2: levels of lipophilic compounds in organic matrices should be lipid normalized; to report lipid content of tissues alongside in the table does not help to interpret the results
282-298: the whole comparison between literature data and results from the present study seems useless, considering that the same authors indicate a difficulty in comparing them based on the differences in the expression of the results (278), the origin of the samples or the analytical methods (299-305)
312-313: feeding on jellyfish, crustaceans, mollusks, and small fish does not make sea turtles real apex predators, since these preys are quite low in the food chain; rather, they are very long-lived organisms and practically devoid of natural predators once they reach an adult size, so they are capable of accumulating persistent pollutants with little/no metabolism/elimination
320-322: these are only hypothesis expressed by the cited authors.
Author Response

(The authors gave the same response as above.)

Round 2
Reviewer 1 Report
Overall, I am satisfied with Authors' replies to my suggestions. I still think that the statistical analyses should be improved, but I can understand the Authors' point of view. I would just read a sentence in the Discussion acknowledging that statistical analyses may be improved.
Author Response
Dear Reviewer,
Thank you for your comment, a few sentences has been added at the end of the discussion to explain that further statistical analyses could be done to improve the study.
Reviewer 3 Report
the last note is for line 120 and/or 507: to be fair, the correct quote is Poppi & Di Bello, or even better Poppi L. & Marchiori E. Dissection Tecniques and Notions, in "Sea turtle management manual", Poppi L. & Di Bello A. Eds.”
Author Response
Dear Reviewer,
Thank you for your comment; we changed the reference as suggested.